# Effects of Transanal Irrigation on Gut Microbiota in Pediatric Patients with Spina Bifida

**DOI:** 10.3390/jcm10020224

**Published:** 2021-01-10

**Authors:** Akira Furuta, Yasuyuki Suzuki, Ryosuke Takahashi, Birte Petersen Jakobsen, Takahiro Kimura, Shin Egawa, Naoki Yoshimura

**Affiliations:** 1Department of Urology, Jikei University School of Medicine, Tokyo 105-8471, Japan; tkimura@jikei.ac.jp (T.K.); s-egpro@jikei.ac.jp (S.E.); 2Department of Urology, Tokyo Metropolitan Rehabilitation Hospital, Tokyo 131-0034, Japan; ysuro@jikei.ac.jp; 3Department of Urology, Spinal Injuries Center, Fukuoka 820-0053, Japan; r-taka@uro.med.kyushu-u.ac.jp; 4Consultant MD, MedDevHealth, Geelsvej, 15 2840 Holte, Denmark; birte@meddevhealth.com; 5Department of Urology, University of Pittsburgh School of Medicine, Pittsburgh, PA 15261, USA; nyos@pitt.edu

**Keywords:** constipation, gut microbiota, spina bifida, transanal irrigation, urinary tract infection

## Abstract

Recent studies using 16S rRNA-based microbiota profiling have demonstrated dysbiosis of gut microbiota in constipated patients. The aim of this study was to investigate the changes in gut microbiota after transanal irrigation (TAI) in patients with spina bifida (SB). A questionnaire on neurogenic bowel disfunction (NBD), Bristol scale, and gut microbiota using 16S rRNA sequencing were completed in 16 SB patients and 10 healthy controls aged 6–17 years. Then, 11 of 16 SB patients with moderate to severe NBD scores received TAI for 3 months. Changes in urine cultures were also examined before and after the TAI treatments. In addition, correlation of gut microbiota and Bristol scale was analyzed. Significantly decreased abundance in *Faecalibacterium*, *Blautia* and *Roseburia*, and significantly increased abundance in *Bacteroides* and *Roseburia* were observed in the SB patients compared with controls and after TAI, respectively. The abundance of *Roseburia* was significantly correlated positively with Bristol scale. Urinary tract infection tended to decrease from 82% to 55% after TAI (*p* = 0.082) despite persistent fecal incontinence. Butyrate-producing bacteria such as *Roseburia* play a regulatory role in the intestinal motility and host immune system, suggesting the effects of TAI on gut microbiota.

## 1. Introduction

Patients with neurogenic diseases affecting the spinal cord such as spina bifida (SB) and spinal cord injury (SCI) often present disturbance of bladder and bowel function. To date, treatments of neurogenic bowel dysfunction (NBD) have been largely empirical, and individual solutions have been sought, whereas clean intermittent catheterization (CIC) has commonly been used to treat neurogenic bladder due to spinal cord lesions. Patients with SB or SCI often suffer from both urinary and bowel symptoms, and expect to undergo the treatments of NBD at the same time. The Peristeen^®^ transanal irrigation system (Coloplast A/S, Humlebaek, Denmark) was for the first time permitted for use in the treatments of intractable neurogenic constipation from March 2018 in Japan. A randomized controlled trial found that SCI patients treated with the Peristeen^®^ transanal irrigation system showed improvements in constipation, fecal incontinence, and symptom-related quality of life compared with patients treated with conservative bowel management as the best supportive bowel care without irrigation [1]. It has also been reported that about 60% (36/60) of SB patients aged 8–17 years showed relief from neurogenic constipation three months after transanal irrigation (TAI) [2].

The human intestinal tract is colonized by hundreds of trillions of bacteria whose number exceeds that of the host cells by ten-fold or more [3]. Gut microbiota act as a barrier against pathogens, stimulate the host immune system, and produce a great variety of compounds from the metabolism of diet that could affect the host [4,5]. Immune function in patients with spinal cord lesions is crucial because of the increased incidence of urinary tract infection (UTI), probably due to the increased residual urine volume [6]. Therefore, the CIC maneuver has been introduced to reduce the incidence of UTI [7]. On the other hand, the growth and composition of gut microbiota are affected by a plethora of factors, including age [8], diet [9], obesity [10], and intestinal motility [11]. Recent studies using 16S rRNA-based microbiota profiling have demonstrated dysbiosis of gut microbiota in constipated patients [11]. However, to the best of our knowledge, there have been no reports concerning the effects of TAI on gut microbiota in constipated patients. Therefore, the aim of this study was to investigate the changes in gut microbiota after TAI in SB patients using 16S rRNA sequencing, which detects microbes that have not yet been cultured but can be assigned as relatives of cultured representatives with known function.

## 2. Experimental Section

### 2.1. Study Design

Sixteen pediatric SB patients aged 6–17 years treated with self or helped CIC due to neurogenic bladder were recruited from the Jikei University Hospital, and 10 age- and sex-matched healthy controls without any disease were included from the same hospital employees’ children between July 2018 and June 2019. SB patients with myelomeningocele at lumbosacral or sacral lesion levels were included. Obese children whose body mass index (BMI) was over 25 kg/m^2^ were excluded. All participants completed the Bristol scale (range 1–7; 1 = separate hard lumps, 4 = like a smooth, soft sausage or snake, 7 = liquid consistency with no solid pieces) and NBD score (range 0–47; 0–6 = very minor, 7–9 = minor, 10–13 = moderate, 14–47 = severe), which consists of 10 questions including frequency of bowel movements (range 0–6; 0 = daily, 6 = less than once a weak), time used for defecation (range 0–7; 0 = 0–30 min, 7 = more than 1 h), headache or perspiration during defecation (range 0–2; 0 = no, 2 = yes), use of tablets against constipation (range 0–2; 0 = no, 2 = yes), use of drops against constipation (range 0–2; 0 = no, 2 = yes), digital stimulation or evacuation (range 0–6; 0 = daily, 6 = less than once a week), frequency of fecal incontinence (range 0–13; 0 = less than once a week, 13 = daily), use of tablets against fecal incontinence (range 0–4; 0 = no, 4 = yes), flatus incontinence (range 0–2; 0 = no, 2 = yes), and perianal skin problems (range 0–3; 0 = no, 3 = yes) in cooperation with their parents.

Eleven of the sixteen intractable constipated SB patients with moderate to severe NBD score then received TAI using the Peristeen^®^ anal irrigation system every two days for 3 months. Changes in urine cultures (10^4^ colony forming units/mL or larger was regarded as bacteriuria) in addition to the Bristol scale and NBD score were also examined before and after the TAI treatments. The use of new gastrointestinal interventions including prebiotics, probiotics, antibiotics, modifications to their diet, or medication for the treatment of constipation were prohibited during the study.

All participants gave their written informed consent to the protocol and were permitted to withdraw from the study at any time. The study was approved by the Ethics Committee of the Jikei University School of Medicine (9156).

### 2.2. Fecal Sample Collection and 16S rRNA Gene Sequencing

Fecal samples were collected from 10 healthy controls without TAI, 16 SB patients before TAI, and 11 SB patients after TAI using dedicated containers (Techno Suruga Laboratory, Shizuoka, Japan) for the analysis of 16S rRNA sequencing and stored at 4 °C in refrigerators until further processing. DNA was isolated with Isospin fecal DNA (Nippon Gene Co., LTD, Tokyo, Japan) according to manufacturer’s instructions.

The exacted DNA was analyzed in Repertoire Genesis Inc. (Osaka, Japan). The V1V2 region of 16S rRNA genes was amplified using FOH-27Fmod (5′-AGRGTTTGATYMTGGCTCAG-3′) and ROH-338R (5′-TGCTGCCTCCCGTAGGAGT-3′) under the following polymerase chain reaction (PCR) conditions: one cycle of denaturation at 95 °C for 3 min, 25 cycles at 95 °C for 30 s, 55 °C for 30 s, 72 °C for 30 s, and final extension at 72 °C for 5 min. The PCR products were sequenced by MiSeq Deep sequencer using MiSeq Reagent Kit v3 (Illumina, San Diego, CA, USA) following the manufacturer’s instructions. The sequence data were preprocessed and analyzed using the “Flora Genesis software” (Repertoire Genesis Inc.). In brief, the R1 and R2 read pairs were joined and chimera sequences were removed. The operational taxonomic unit (OTU) picking was performed by the open-reference method using the 97% identity prefiltered Greengenes database and the UCLUST. The representative sequences of each OTU were chosen and taxonomy assignment was performed by Ribosomal Database Project (RDP) classifier using a threshold score of 0.5 or more. The OTUs were grouped if their annotation was the same regardless of their RDP score.

### 2.3. Statistical Analysis

All data were represented as mean values ± standard deviation of the mean. Statistical analysis software (Prism, GraphPad Software, San Diego, CA, USA) was used to perform the data analysis.

Significant differences in the age, BMI, Bristol scale, and NBD score in the controls and SB patients were analyzed using Mann–Whitney U-test, whereas the sex differences were detected by Chi-square test. Meanwhile, changes in the Bristol scale, NBD score, and UTI after TAI in the SB patients were examined using Wilcoxon signed rank test.

All microbes at the phylum level were compared, whereas the microbes at the genus level with low relative abundances (<0.1%) were filtered and the remaining top 50 different types were analyzed. Significant differences in the relative abundance of microbes in the controls and SB patients were detected using Mann–Whitney U-test. Changes in the relative abundance of microbes after TAI in the SB patients were examined by Wilcoxon signed rank test. In addition, the correlation of the microbes and Bristol scale was analyzed by Spearman rank correlation.

## 3. Results

### 3.1. Characteristics in the SB Patients Compared with Healthy Controls

There were not significant differences in the sex, age, and BMI between the controls and SB patients, as shown in Table 1. On the other hand, the Bristol scale was significantly decreased and the NBD score was significantly increased in the SB patients compared with the controls. All of the SB patients had mild (3), moderate (4), or severe (9) neurogenic constipation.

### 3.2. Comparison of Gut Microbiota in the SB Patients and Healthy Controls

At the phylum level, there were no significant differences in the relative abundance of microbes in the controls and SB patients, although a total of 15 microbes were detected (Figure 1A, Table 2). On the other hand, at the genus level, the relative abundance of *Faecalibacterium*, *Blautia*, *Roseburia*, *Lachnospira*, and *Dialister* was significantly decreased and that of *Oscillospira* was significantly increased in the SB patients before TAI compared with the controls (Figure 1B, Table 2).

### 3.3. Outcome Measures before and after TAI in the SB Patients

Significant changes in the Bristol scale and total NBD score were observed after the TAI treatments in the constipated SB patients. The results analyzing the sub-NBD score showed that use of tablets against constipation was significantly decreased without significant changes in the frequency of fecal incontinence. Asymptomatic bacteriuria caused by *Escherichia coli* tended to decrease from 82% to 55% after TAI, although the difference was not statistically significant (*p* = 0.082) (Table 3).

### 3.4. Changes in Gut Microbiota after TAI in the SB Patients

At the phylum level, there were no significant changes in the relative abundance of 15 microbes after the TAI treatments (Figure 2A, Table 4). At the genus level, the relative abundance of *Bacteroides* and *Roseburia* was significantly increased and that of *Turicibacter* was significantly decreased after TAI in SB patients (Figure 2B, Table 4).

### 3.5. Correlation of Gut Microbiota and Bristol Scale

The relative abundance of *Roseburia* was significantly correlated positively with the Bristol scale, although a significant correlation was not detected in *Bacteroides, Faecalibacterium*, or *Blautia* (Table 5).

## 4. Discussion

The results of this study indicate that Bristol scale and total NBD score were significantly deteriorated in the SB patients compared with healthy controls and then significantly improved after TAI in the SB patients, which is consistent with previous reports [1,2] It was also observed that 82% of the SB patients had asymptomatic UTI predominantly caused by *E. coli* before TAI, which then tended to decrease after TAI (*p* = 0.082). Therefore, it is assumed that TAI, by improving bowel habit and washing of the colorectal tract, can reduce the risk of bladder contamination by *E. coli* [2], although persistent fecal incontinence after TAI due to atonic sphincters remained after TAI in this study.

An innovative point of the present study was to investigate the changes in gut microbiota before and after TAI in the constipated SB patients, with the exclusion of probable confounding effects of age, diet, and intestinal motility on gut microbiota. In addition, we decided to enroll non-obese patients with a body mass index of 25 kg/m^2^ or less because obesity has a large impact on gut microbial composition (based on a previous report demonstrating that an increased *Firmicutes*/*Bacteroidetes* ratio is associated with obesity) [10].

*Firmicutes* and *Bacteroidetes* were the most predominant bacteria at the phylum level in the gut, in accordance with a previous report [12]. They ferment indigestible carbohydrates and generate short-chain fatty acids (SCFAs) including acetate, propionate, and butyrate. It has been reported that *Bacteroidetes* produce high levels of acetate and propionate, whereas *Firmicutes* produce high amounts of butyrate [13]. The most numerous butyrate-producing bacteria are highly oxygen-sensitive anaerobes belonging to the *Clostridial* clusters IV, including *Faecalibacterium*, and XIVa, including *Roseburia* and *Lachnospira* [14,15]. The abundance of *Faecalibacterium* has significantly been decreased in patients with functional constipation [16]. Significantly decreased abundance in *Roseburia* was also observed in patients with functional constipation [16,17] or in constipated patients with irritable bowel syndrome (IBS) as well as in SCI patients compared with healthy controls [18,19,20]. In addition, *Blautia* or *Dialister* has been reported to produce acetate or propionate, respectively [21,22]. In the present study, the abundance of *Blautia* and *Dialister* as well as *Faecalibacterium, Roseburia*, and *Lachnospira* was significantly decreased in the SB patients.

Gut microbiota obtained from constipated patients with IBS were observed to produce more sulfides and hydrogen and less butylate from starch fermentation than healthy controls [18]. This study showed that the abundance of *Roseburia* was significantly correlated positively with Bristol scale, which is possibly because butyrate plays a regulatory role in the transepithelial fluid transport and intestinal motility via release of 5-hydroxytryptamine [23]. On the other hand, the anti-inflammatory effects of SCFAs are mediated through binding of the G-protein-coupled receptor 41 and 43, which are both expressed on immune cells, suggesting that SCFAs are involved in the activation of leucocytes [23]. Significantly decreased levels of butylate and acetate has been previously reported in patients with irritable bowel diseases compared with health controls [24]. In the present study, significantly increased abundance in *Roseburia and Bacteroides* was observed after TAI, which may contribute to the tendency for UTIs to be reduced in the constipated SB patients treated with CIC despite the persistent fecal incontinence remaining. Furthermore, as mentioned above, significantly increased abundance in *Bacteroides* was observed after TAI in this study, whereas significantly decreased abundance in *Bacteroides* and *Prevotella* has been reported in constipated patients with IBS [25]. Three enterotypes, including *Bacteroides*, *Prevotella*, and *Ruminococcus* at the genus level, have recently been defined in a global collection of gut microbiota [26]. The results of this study therefore suggested the contribution of TAI to the changes in *Bacteroides*, one of the enterotypes.

It has been reported that *Oscillospira* is closely related to human health because its abundance is negatively correlated with systolic and diastolic blood pressure, fasting blood glucose, triglyceride, uric acid, and Bristol scale, suggesting that *Oscillospira* is a predictor of low BMI and constipation [27]. In addition, an association between *Turicibacter* and exercise has been reported in mice. A previous study showed that the percentage of *Turicibacter* in controls was 0.22%, whereas that of *Turicibacter* in voluntary wheel running groups was 0% [28]. In the present study, significantly decreased abundance in *Turicibacter* was observed after TAI, suggesting the recovery of intestinal motility possibly due to butyrate in the SB patients.

Psyllium is widely used for the treatment of constipation. It is capable of retaining water in the small intestine, and thereby increasing water flow into the ascending colon. The increases in the fluidity of colonic content may explain the success of psyllium in treating constipation [29]. Interestingly, psyllium supplementation increased fecal water, resulting in the significant increases in *Faecalibacterium, Roseburia*, and *Lachnospira* in the patients with functional constipation [30]. Similarly, the present study for the first time suggested the effects of TAI on gut microbiota, especially butyrate-producing bacteria such as *Roseburia*.

A major limitation of this study was the small number of patients, but confounding effects of age, diet, obesity, and intestinal motility on gut microbiota were excluded by comparing gut microbial composition before and after the TAI treatment. In addition, we only characterized gut microbiota without investigating their metabolites, such as SCFAs. Further studies are needed to clarify this point.

## 5. Conclusions

TAI significantly improved constipation in addition to significantly increasing the abundance in *Roseburia*, which may contribute to improvements in the host immune system, resulting in the tendency for UTIs to be reduced, despite persistent fecal incontinence. Therefore, TAI combined with CIC could be beneficial for improving bowel dysfunction in constipated patients with spinal cord lesions such as SB.

## Figures and Tables

**Figure 1 jcm-10-00224-f001:**
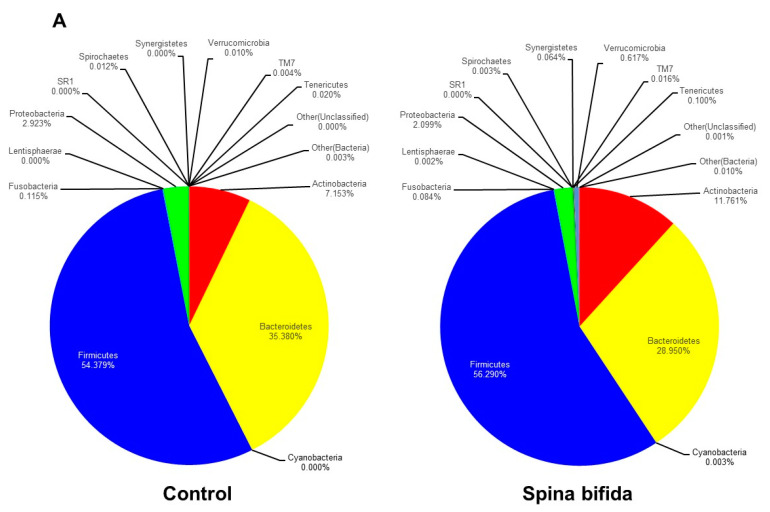
Comparison of gut microbiota in the spina bifida (SB) patients and healthy controls at the phylum level (**A**) and the genus level (**B**). *; *p* < 0.05, **; *p* < 0.01 vs. Control.

**Figure 2 jcm-10-00224-f002:**
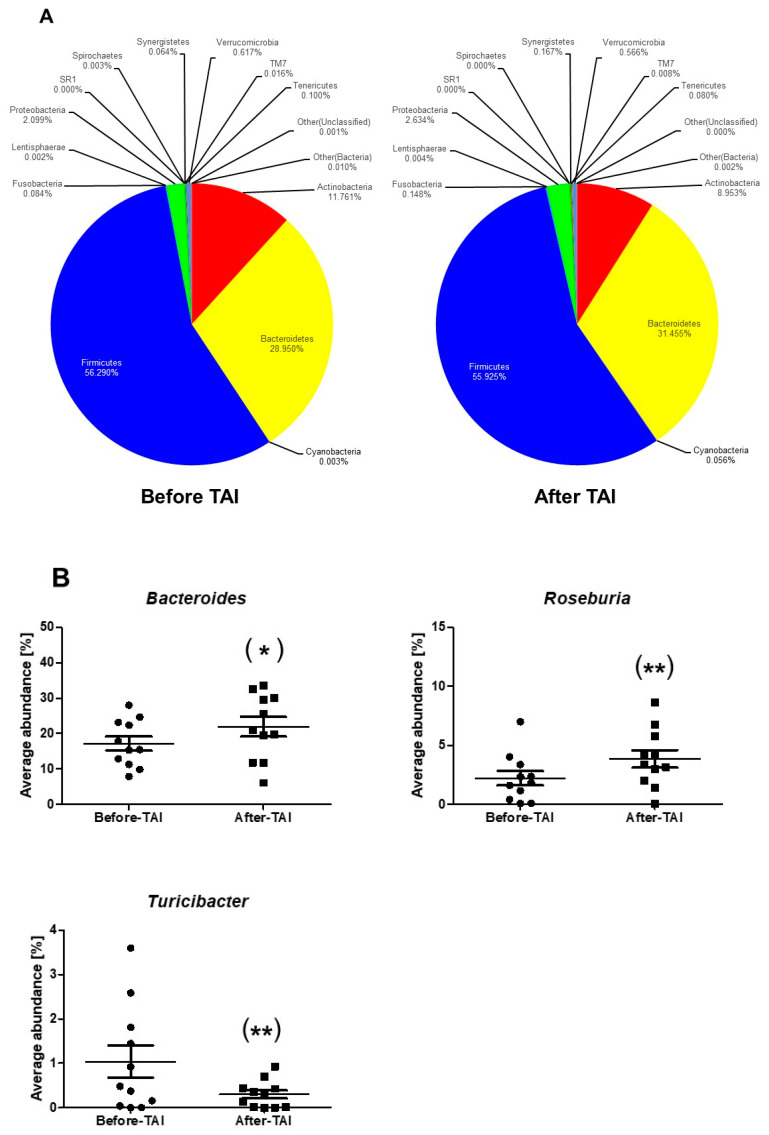
Changes in gut microbiota after TAI in the SB patients at the phylum level (**A**) and the genus level (**B**). *; *p* < 0.05, **; *p* < 0.01 vs. before TAI.

**Table 1 jcm-10-00224-t001:** Characteristics of the study groups. BMI: body mass index; CIC: clean intermittent catheterization; NBD: neurogenic bowel dysfunction.

	Healthy Control	Spina Bifida	*p* Value
Participants (Number)	10	16	
Male/Female (Number)	5/5	5/11	0.339
Age (Mean ± SD )	12.6 ± 2.5	10.8 ± 3.3	0.223
BMI (Mean ± SD )	17.5 ± 1.6	18.9 ± 3.8	0.429
CIC (Number)	0	16	
Bristol scale (Mean ± SD )	3.8 ± 0.4	1.8 ± 1.0	0.001
NBD score (Mean ± SD )	0.5 ± 1.0	14.1 ± 4.7	0.001
Very mild (Number)	10	0	
Mild (Number)	0	3	
Moderate (Number)	0	4	
Severe (Number)	0	9	

**Table 2 jcm-10-00224-t002:** Differences between gut microbiota in the SB patients and healthy controls.

Phylum Level			
Family Level	Control	Spina Bifida	*p* Value
Genus Level	(Mean)	(Mean)	
Firmicutes/Bacteroidetes ratio	1.58	2.38	0.246
Firmicutes	54.38	56.29	0.654
Ruminococcaceae			
* Ruminococcus*	2.79	6.30	0.108
* Faecalibacterium*	10.64	6.33	0.033
* Oscillospira*	0.98	1.74	0.029
Lachospiraceae			
* Blautia*	9.00	5.38	0.017
* Ruminococcus*	8.44	8.53	0.693
* Roseburia*	4.99	2.06	0.001
* Dorea*	1.71	1.73	0.937
* Lachnospira*	1.15	0.24	0.002
* Coprococcus*	1.07	0.55	0.654
* Lactobacillus*	0.03	1.18	0.346
Streptococcaceae			
* Streptococcus*	1.98	1.82	0.304
Clostridiaceae			
* SMB53*	1.15	2.50	0.120
Veillonellaceae			
* Veillonella*	1.22	0.97	0.051
* Dialister*	1.20	0.22	0.003
Bacillaceae			
* Bacillus*	0.04	1.89	0.593
Erysipelotrichaceae			
* Eubacterium*	0.20	0.61	0.055
Turicibacteraceae			
* Turicibacter*	0.24	0.80	0.144
Bacteroidetes	35.38	28.95	0.147
Bacteroidaceae			
* Bacteroides*	27.66	20.11	0.087
Porphyromonadaceae			
* Parabacteroides*	2.34	3.69	0.257
Prevotellaceae			
* Prevotella*	3.05	0.06	0.477
Actinobacteria	7.15	11.76	0.133
Bifidobacteriaceae			
* Bifidobacterium*	5.89	8.67	0.280
Coriobacteriaceae			
* Collinsella*	0.96	2.01	0.684
Proteobacteria	2.92	2.09	0.414
Alcaligenaceae			
* Sutterella*	1.81	1.07	0.087
Enterobacteriaceae			
* Trabulsiella*	0.36	0.41	0.385

**Table 3 jcm-10-00224-t003:** Influence of transanal irrigation (TAI) on outcome measures.

	Before TAI	After TAI	*p* Value
Bristol scale ( Mean ± SD )	1.9 ± 1.2	3.6 ± 1.2	0.001
Total NBD score ( Mean ± SD )	15.6 ± 4.1	11.1 ± 4.6	0.009
Frequency of bowel movements ( Mean ± SD )	1.7 ± 2.4	1.0 ± 0.0	0.279
Time used for defecation ( Mean ± SD )	0.9 ± 2.1	1.1 ± 1.5	0.828
Headache or perspiration during defecation ( Mean ± SD )	0.6 ± 0.9	0.1 ± 0.5	0.082
Use of tablets against constipation ( Mean ± SD )	1.1 ± 1.0	0.4 ± 0.9	0.019
Use of drops against constipation ( Mean ± SD )	0.4 ± 0.9	0.0 ± 0.0	0.082
Digital stimulation or evacuation ( Mean ± SD )	4.3 ± 2.8	3.4 ± 3.1	0.336
Frequency of fecal incontinence ( Mean ± SD )	5.0 ± 3.7	3.7 ± 3.4	0.108
Use of tablets against fecal incontinence ( Mean ± SD )	0.3 ± 1.1	0.0 ± 0.0	0.336
Flatus incontinence ( Mean ± SD )	2.0 ± 0.0	2.0 ± 0.0	1.000
Perianal skin problems ( Mean ± SD )	0.9 ± 1.4	0.2 ± 0.8	0.083
Urinary tract infection (Number, %)	9 (82%)	6 (55%)	0.082
Causative bacteria (Number) *Escherichia coli*	9	6	

**Table 4 jcm-10-00224-t004:** Changes in gut microbiota after TAI in the SB patients.

Phylum Level			
Family Level	Before TAI	After TAI	*p* Value
Genus Level	(Mean)	(Mean)	
Firmicutes/Bacteroidetes ratio	2.53	2.36	0.638
Firmicutes	58.57	57.28	0.638
Ruminococcaceae			
* Ruminococcus*	7.48	7.48	1.000
* Faecalibacterium*	7.30	9.27	0.320
* Oscillospira*	1.72	1.83	0.700
Lachospiraceae			
* Blautia*	6.11	7.45	0.240
* Ruminococcus*	8.70	8.05	0.966
* Roseburia*	2.22	3.86	0.007
* Dorea*	2.28	2.12	0.638
* Coprococcus*	0.75	0.54	1.000
Streptococcaceae			
* Streptococcus*	1.49	2.00	0.700
Clostridiaceae			
* SMB53*	2.98	1.00	0.067
* 02d06*	0.90	0.53	0.695
Veillonellaceae			
* Phascolarctobacterium*	0.50	0.47	0.938
Erysipelotrichaceae			
* Eubacterium*	0.74	0.83	0.898
Turicibacteraceae			
* Turicibacter*	1.04	0.30	0.003
Bacteroidetes	28.92	31.65	0.638
Bacteroidaceae			
* Bacteroides*	17.35	21.92	0.048
Porphyromonadaceae			
* Parabacteroides*	3.03	3.34	0.831
Prevotellaceae			
* Prevotella*	1.96	2.33	0.557
Odoribacteraceae			
* Odoribacter*	0.51	0.49	0.922
Actinobacteria	9.74	7.67	0.320
Bifidobacteriaceae			
* Bifidobacterium*	6.79	5.93	0.413
Coriobacteriaceae			
* Collinsella*	1.65	1.43	0.250
Proteobacteria	1.84	2.52	0.175
Alcaligenaceae			
* Sutterella*	0.71	1.35	0.059

**Table 5 jcm-10-00224-t005:** Correlation of gut microbiota and Bristol scale.

Bacteria	Bristol Scale	*p* Value
(Genus level)	(Correlation coefficient)	
*Bacteroides*	0.262	0.147
*Faecalibacterium*	0.239	0.187
*Blautia*	0.264	0.144
*Roseburia*	0.486	0.005

## Data Availability

The data that support the findings are available from the corresponding author (A.F.) upon reasonable request.

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
