# Peer review of "Effects of Transanal Irrigation on Gut Microbiota in Pediatric Patients with Spina Bifida"

_jcm, 2021, doi:10.3390/jcm10020224_

Round 1

Reviewer 1 Report

The aim of the study was to investigate the changes in gut microbiota after transanal irrigation in patients with spina bifida using 16S rRNA sequencing. The topic is original and conducted in a very specific population. However I have some limitations, mainly regarding the methodological part of the study.

INTRODUCTION

  • As there ara data available in literature focusing especially on NBD in patients with spina bifida, the authors could develop it a little more in the introduction.

METHOD

  • Dates of the inclusion of patients are not specified.
  • There is a bias in the inclusion of hospital employee's children which is not a representative sample of HV. This could be noted in the discussion.
  • No information is given regarding the HV group (criteria of inclusion and exclusion). Could the authors bring more details?
  • The NBD score can be difficult to apply in a pediatric population (in particular with young children). The authors did not precise the reference they used. Did they use a pediatric version that was developed in a population of children with SB (doi 10.1002/nau.22694 )? If they used the adult version of the questionnaire, could they detail more how the children filled the questionnaire and if they were helped. A stool calendar could also have been appropriated in this study.
  • There is a lack of information regarding the type of spina bifida in patients, could the authors precise it?
  • A few informations are given regarding the voiding mode: did patients perform CIC on a mitrofanoff or via natural ways? self or helped CIC? Were they treated for neurogenic bladder and they had one, if yes could the authors precise? It could influence both bowel and bladder control.
  • What is the definition of asymptomatic UTI found in the results part?
  • Had the SB patients gastrointestinal explorations before the use of TAI? Could they continue laxative medication during the follow-up?
  • A limit in the method is that no information is given on potential bias on gut microbiote at baseline and during the 3-months period. Could patients take medication like probiotics, antibiotics or modified their diet? Did they had UTI? or gastro intestinal infection? Was the use of new gastrointestinal medication permitted? Did the authors assessed these counfonding factors?
  • As the authors used non parametric tests in the statistical analysis, the results should be given as median +/- IQ rather than mean +/- SD.

RESULTS

  • The authors could add in Table 1 informations asked in comments of the Method part.
  • Could the authors put in bold the significant p<0.05 and homogenize to 2 digits after the decimal point in the Tables? It would facilitate the reading of the Tables.
  • 3.3: what did the author mean with "UTI caused by Escherichia coli"? Was it symptomatic or asymptomatic UTI?

DISCUSSION

  • In the sentence line 191 "An innovative point of the present study was to investigate the changes ... confounding effects of age, diet and intestinal motility on gut microbiot". No information regarding diet of both controls and patients is given in the study. Could the authors precise it? If the diet was assessed it should be noted in the Method part of the manuscript.

CONCLUSION

  • In the sentence "Therefore, TAI combined with CIC would be beneficial for improving bowel and urinary dysfunction in the constipated patients with spinal cord lesions such as SB" . The authors should suppress "urinary function" as no complete evaluation of the urinary tract was performed in the study.

Author Response

INTRODUCTION:

Q1: As there are data available in literature focusing especially on NBD in patients with spina bifida, the authors could develop it a little more in the introduction.

A1: According to the comment, we added “It has also been reported that about 60% (36/60) of SB patients aged 8-17 years showed the relief from neurogenic constipation three months after transanal irrigation (TAI) [2].” P1, Line 43-44.

METHOD:

Q2: Dates of the inclusion of patients are not specified.

A2: Thank you for the comment; we added “between July 2018 and June 2019”. P2, Line 66-67.

Q3: There is a bias in the inclusion of hospital employee's children which is not a representative sample of HV. This could be noted in the discussion.

No information is given regarding the HV group (criteria of inclusion and exclusion). Could the authors bring more details?

A3: Thank you for the comment; we revised “age and sex-matched 10 healthy controls were included from the same hospital employees’ children” to “age and sex-matched 10 healthy controls were included from the same hospital employees’ children without any disease”. P2, Line 565-66.

In addition, we think that the hospital employees’ (clinical doctors’) children are a representative sample of controls.

Q4: The NBD score can be difficult to apply in a pediatric population (in particular with young children). The authors did not precise the reference they used. Did they use a pediatric version that was developed in a population of children with SB (doi 10.1002/nau.22694 )? If they used the adult version of the questionnaire, could they detail more how the children filled the questionnaire and if they were helped. A stool calendar could also have been appropriated in this study.

A4: Thank you for the important comment; we added “in cooperation with their parents”. P2, Line 78.

Q5: There is a lack of information regarding the type of spina bifida in patients, could the authors precise it?

A5: Thank you for the comment; we added “SB patients with myelomeningocele at lumbosacral or sacral lesion levels were included.” P2, Line 67-68.

Q6: A few information are given regarding the voiding mode: did patients perform CIC on a mitrofanoff or via natural ways? self or helped CIC? Were they treated for neurogenic bladder and they had one, if yes could the authors precise? It could influence both bowel and bladder control.

A6: According to the comment, we revised “Sixteen pediatric SB aged 6-17 years treated with CIC were recruited” to “Sixteen pediatric SB patients aged 6-17 years treated with self or helped CIC due to neurogenic bladder were recruited”. P2, Line 64-65.

Q7: What is the definition of asymptomatic UTI found in the results part?

A7: Thank you for the comment; we revised “UTI caused by Escherichia coli tended to decrease from 82% to 55% after TAI” to “Asymptomatic bacteriuria caused by Escherichia coli tended to decrease from 82% to 55%”. P5, Line 171-172.

We also revised “Changes in urine cultures (104 colony forming unit/mL or larger was regarded as positive)” to “Changes in urine cultures (104 colony forming unit/mL or larger was regarded as bacteriuria)”. P2, Line 80-81.

Q8: Had the SB patients gastrointestinal explorations before the use of TAI? Could they continue laxative medication during the follow-up?

A8: Thank you for the comment; we did not perform the bowel functional explorations before the use of TAI. The results analyzing the sub-NBD score showed that use of tablets against constipation was significantly decreased in the study. P5, Line 169-170.

Q9: A limit in the method is that no information is given on potential bias on gut microbiota at baseline and during the 3-months period. Could patients take medication like probiotics, antibiotics or modified their diet? Did they have UTI? or gastro intestinal infection? Was the use of new gastrointestinal medication permitted? Did the authors assess these confounding factors?

A9: Thank you for the important comment; we added “The use of new gastrointestinal interventions including prebiotics, probiotics, antibiotics, modified their diet or medication for the treatment of constipation was prohibited during the study.” P2, Line 82-84.

Q10: As the authors used non parametric tests in the statistical analysis, the results should be given as median +/- IQ rather than mean +/- SD.

A10: Your advice is appropriate, but the data represented as mean ± SD are commonly used and easy to understand the results. We described “All data were represented as mean values ± standard deviation of the mean.” P3, Line 116.

RESULTS:

Q11: The authors could add in Table 1 information asked in comments of the Method part.

A11: According to the comment, we added the aforementioned sentences in the text, but not in Table 1.

Q12: Could the authors put in bold the significant p<0.05 and homogenize to 2 digits after the decimal point in the Tables? It would facilitate the reading of the Tables.

A12: According to the comment, we put in bold the significant p<0.05 in Table 1-5. On the other hand, the representation of 3 digits after the decimal point remained because p=0.001 is considered to be meaningful compared with p<0.01.

Q13: 3.3: what did the author mean with "UTI caused by Escherichia coli"? Was it symptomatic or asymptomatic UTI?

A13: Thank you for the comment; we revised “UTI caused by Escherichia coli tended to decrease from 82% to 55% after TAI” to “Asymptomatic bacteriuria caused by Escherichia coli tended to decrease from 82% to 55%” as shown in A7. P5, Line 171-172.

DISCUSSION:

Q14: In the sentence line 191 "An innovative point of the present study was to investigate the changes ... confounding effects of age, diet and intestinal motility on gut microbiota". No information regarding diet of both controls and patients is given in the study. Could the authors precise it? If the diet was assessed it should be noted in the Method part of the manuscript.

A14: According to the comment, we added “The use of new gastrointestinal interventions including prebiotics, probiotics, antibiotics, modified their diet or medication for the treatment of constipation was prohibited during the study.” as shown in A9. P2, Line 82-84.

CONCLUSION:

Q15: In the sentence "Therefore, TAI combined with CIC would be beneficial for improving bowel and urinary dysfunction in the constipated patients with spinal cord lesions such as SB". The authors should suppress "urinary function" as no complete evaluation of the urinary tract was performed in the study.

A15: Thank you for the comment; we revised it to “Therefore, TAI combined with CIC would be beneficial for improving bowel dysfunction in the constipated patients with spinal cord lesions such as SB.” P9, Line 279-281.

Reviewer 2 Report

Dear Authors,

I congratulate with your work. The topic is of main interest. The methodology employed is good. The article is easy to be read. 

Some minor comments: 

In line 40 you state that CIC is the gold standard for the bladder management in SCI patients: it is not always true, as bladder management (CIC, reflex micturitions, permanent drainage...)  depends on the lesional level, on the completeness of the spinal cord injury and on the bladder functional status (compliance, capacity...).

In line 41 you state that patients using CIC often suffer from constipation, fecal incontinence or both: I would state that patients with SB or SCI ofen suffer from bowel symptoms, CIC maneuver do not have direct bowel implications.

In line 56 you state that the main cause of UTI is probably CIC: the CIC maneuver actually has been introduced to reduce the incidence of UTI in neurogenic patients. As you demontrated in your study gut microbiota change had an impact on UTI incidence (although not significative), in a population continuing CIC maneuver. Probably gut microbiota may have a much bigger impact that a correct CIC maneuver itself.  

Was the indication to TAI given only with the use of the Bristol scale and NDB score? Didn't you perform any additional test? (such as Rx-intenstinal transit with radiopaque markers or others) 

Do you think that the incidence of fecal incontinence after TAI may also be due to an inappropriate use of the device (too much solution injected, short time for evacuation...)?

Author Response

Q1: In line 40 you state that CIC is the gold standard for the bladder management in SCI patients: it is not always true, as bladder management (CIC, reflex micturitions, permanent drainage...)  depends on the lesional level, on the completeness of the spinal cord injury and on the bladder functional status (compliance, capacity...).

A1. According to the comment, we revised it to “clean intermittent catheterization (CIC) has commonly been used to treat neurogenic bladder due to spinal cord lesions”. P1, Line 35-36.

Q2: In line 41 you state that patients using CIC often suffer from constipation, fecal incontinence or both: I would state that patients with SB or SCI often suffer from bowel symptoms, CIC maneuver do not have direct bowel implications.

A2. According to the comment, we revised it to “The patients with SB or SCI often suffer from both urinary and bowel symptoms, and expect to undergo the treatments of NBD at the same time.” P1, Line 36-37.

Q3: In line 56 you state that the main cause of UTI is probably CIC: the CIC maneuver actually has been introduced to reduce the incidence of UTI in neurogenic patients. As you demonstrated in your study gut microbiota change had an impact on UTI incidence (although not significative), in a population continuing CIC maneuver. Probably gut microbiota may have a much bigger impact that a correct CIC maneuver itself.

A3: Thank you for the comment; we revised it to “Immune function in patients with spinal cord lesions is crucial because of the increased incidence of urinary tract infection (UTI) probably due to the increased residual urine volume [5]. Therefore, CIC maneuver has been introduced to reduce the incidence of UTI [6]”. P2, Line 51-53.

In Discussion, we also revised “82% of the SB patients had asymptomatic UTI predominantly caused by E. coli before TAI possibly due to CIC for the treatment of neurogenic bladder, which then tended to decrease after TAI (p=0.082).” to “82% of the SB patients had asymptomatic UTI predominantly caused by E. coli before TAI, which then tended to decrease after TAI (p=0.082).” P8, Line 214-216.

Q4: Was the indication to TAI given only with the use of the Bristol scale and NBD score? Didn't you perform any additional test? (such as Rx-intestinal transit with radiopaque markers or others)

A4: Thank you for the comment; we did not perform the bowel functional explorations. The indication of TAI is “intractable constipation due to spinal cord lesions” in Japan. We revised “The 11 of 16 SB patients with moderate to severe NBD score then received TAI” to “The 11 of 16 intractable constipated SB patients with moderate to severe NBD score then received TAI”. P2, Line 79.

Q5: Do you think that the incidence of fecal incontinence after TAI may also be due to an inappropriate use of the device (too much solution injected, short time for evacuation...)?

A5: Thank you for the comment; The incidence of fecal incontinence after TAI may also be due to an inappropriate use of the device although we instructed the use of device every outpatient visit.

Round 2

Reviewer 1 Report

The authors quickly provided answers to all questions and improved their manuscript.